# Numerical Investigation on the Effect of Height-to-Radius Ratio on Flow Separation Features in S-Shaped Diffuser with Boundary Layer Ingestion

Zhiping Li, Yujiang Lu, Tianyu Pan * and Yafei Zhang

Research Institute of Aero-Engine, Beihang University, Beijing 100191, China; leezip@buaa.edu.cn (Z.L.); luyujiang5464@buaa.edu.cn (Y.L.); bhu12041234@buaa.edu.cn (Y.Z.)
* Correspondence: pantianyu@buaa.edu.cn

**Abstract:** The flow separation occurring in the S-shaped diffuser with boundary layer ingestion (BLI) has a significant effect on the performance of the embedded engine. Previous studies have found that the area ratio (AR) as well as the length-to-offset ratio (LOR) of the S-shaped diffuser are the key contributing factors that affect the flow separation features. Based on the flow phenomena observed in previous studies of an S-shaped diffuser with 100% BLI, a hypothesis that the parameter height-to-radius ratio (HRR) may also have significant effect on the flow separation features in the S-shaped diffuser is proposed. The purpose of this paper is to verify this hypothesis and to further investigate the effect of HRR on the flow separation features in the S-shaped diffuser with BLI using numerical methods. First, the hypothesis that HRR has an effect on the flow separation features in the S-shaped diffuser is verified under uniform inlet condition. Second, the effect of HRR on the flow separation features is investigated under different relative heights of inlet BLI. It is found that the flow separation features in the S-shaped diffuser are very sensitive to the change in HRR but not to the change in relative height of inlet BLI. Finally, for the fixed boundary layer height generated from the airframe, the S-shaped diffuser with a smaller design HRR can significantly suppress the flow separation and thus achieve a higher total pressure recovery and a lower distortion coefficient. The results provide improved understandings of the factor affecting the flow separation features in the S-shaped diffuser, and are useful for improving the aerodynamic performance of the embedded engine with BLI.

**Keywords:** S-shaped diffuser; boundary layer ingestion; flow separation features; height-to-radius ratio

## 1. Introduction

The desire to produce an aerodynamically efficient aircraft led designers to develop new aircraft concepts [1–3]. The representative blended-wing-body (BWB) aircraft with distributed embedded engines provides a unique idea for the next generation of aircraft design, with great potential in reducing power consumption, noise and emissions [4,5]. To hold promise for realizing these benefits, boundary layer ingestion (BLI) is an important technology for such propulsion airframe integration. The benefit of BLI comes from lower inlet velocity, which enables the embedded engine to generate same thrust with less kinetic energy to be wasted compared to conventional podded engines [6,7].

For this configuration, the boundary layer flow developed along the upper surface of the BWB aircraft is ingested into the intake of the embedded engine, and such intake often takes S-shaped geometry due to the longitudinal offset between the inlet axis and the fan axis. Applying an S-shaped diffuser has great advantage in compact engine designs and weight savings, which is different from the conventional straight ducts [8]. Well-designed S-shaped diffusers should efficiently decelerate the incoming flow. To realize the power-saving benefit of BLI for the embedded engine configuration, the S-shaped diffuser must also produce minimal total pressure losses and deliver nearly uniform flow

at the downstream fan entrance [9]. However, this poses great challenges for the S-shaped diffuser design, especially when a large amount of boundary layer flow is ingested into the diffuser [10]. The main reason is that the low-momentum boundary layer flow makes flow separation likely, resulting in severe total pressure losses and distortion at the downstream fan entrance [11,12]. Therefore, the key to improving the performance of embedded engines is to identify the causes of the flow separation in the S-shaped diffuser and to utilize some approaches to suppress flow separation in design procedures.

Previous studies have found that the flow separation in the S-shaped diffuser is generated by the continuous deceleration of the boundary layer due to the adverse pressure gradient near the inner wall [13,14]. This adverse pressure gradient is caused by a combination of flow diffusion and flow turning [15,16]. Based on this understanding, researchers have summarized two global parameters, area ratio (AR) and length-to-offset ratio (LOR), which are the key contributing factors that affect the flow separation features in the S-shaped diffuser [17]. The AR, which is the ratio of the outlet to inlet areas, is the contributing factor of flow diffusion. The LOR, which is the ratio of the length to offset, describes the curvature of the duct and is the contributing factor of flow turning. In addition, more detailed design parameters such as centerline geometry and area distribution of cross-sections can also affect the flow separation features, but these effects should be discussed under fixed global parameters. When the global parameters of the S-shaped diffuser are set, the scale of its internal flow separation is basically determined. In order to suppress this flow separation to achieve lower total pressure losses and a more uniform outlet flow field, researchers have used different technical approaches, which can be classified as flow control methods and shape optimization methods.

For the flow control methods, Gorton et al. [18] conducted a series of experiments to compare the differences between active flow control and passive flow control. The results showed that a measured reduction in total pressure distortion from 29% to 4.6% is achieved by using the pulsed jet with less than 1% of inlet mass flow. Owens et al. [19,20] evaluated the effectiveness of active flow control in reducing total pressure distortion at the fan entrance. The active flow control devices are distributed jets using high-mass flow actuators. The results showed that the circumferential distortion can be reduced from 5.5% to 1.5% using 2.5% of inlet mass flow. Anabtawi et al. [21] used vortex generators to control the flow structure in an S-shaped diffuser. Through different combinations of vortex generators with various configurations, the total pressure distortion can be decreased by about 11%, but an additional total pressure loss is also introduced. For the shape optimization methods, Chiang et al. [22] proposed a high-fidelity aerodynamic shape optimization framework for the optimization of a boundary layer-ingesting S-shaped diffuser. The results indicated that compared to the baseline geometry, a simultaneous improvement in all objectives contained in the composite objective function can be obtained. Lee et al. [23] used the discrete adjoint method to optimize the shape of surface geometry at the intake entrance. The inlet-floor shape is parameterized with the control points on B-spline surface patches, and the results showed that a more than 50% reduction in the flow distortion and a 3% increase in total pressure is reached. Rodriguez et al. [24,25] established a multidisciplinary optimization design method for BLI inlets. The nonlinear optimizer is integrated with an aerodynamics analysis method and a propulsion system simulator. The results showed obvious improvement in pressure recovery and flow uniformity.

These previous works have had varying degrees of success in suppressing the flow separation to improve the performance of S-shaped diffuser. From a system perspective, the purpose of these efforts is to improve the performance of the S-shaped diffuser and the downstream fan by suppressing flow separation due to BLI, and ultimately to maximize the power-saving benefit of BLI as much as possible. To achieve the same goal, the authors attempted to modify the configuration of the embedded engine to solve this problem. In the previous work, a layered embedded engine (LEE) concept, in which the freestream and the boundary layer flow are ingested separately to improve the power-saving benefit of BLI, is proposed. In the design procedure of this new configuration, it is found that when

the boundary layer flow is completely ingested into an S-shaped diffuser with a smaller spacing between the inner and outer walls, the internal flow separation can be significantly suppressed. This phenomenon motivated us to conduct further research, and the purpose of this paper is to provide an explanation for such a phenomenon. First, a hypothesis that the parameter height-to-radius ratio (HRR) may have significant effect on the flow separation features in the S-shaped diffuser is proposed. Second, numerical methods are used to verify this hypothesis and the effect of HRR on the flow separation features in the S-shaped diffuser with BLI is further investigated. Finally, the possible applications of the findings of this study in practical design are discussed.

This paper is organized as follows. The problem descriptions and hypothesis are proposed in Section 2. Then, the methodology including the study subject, numerical methods, validation and test cases are introduced in Section 3. Next, in Section 4, the verification of the hypothesis under uniform inlet condition is proposed. The effect of HRR on flow separation features in the S-shaped diffuser with BLI is analyzed in detail, which is followed by the applications under fixed boundary layer thickness. Finally, the main findings of this paper are concluded in Section 5.

## 2. Problem Descriptions and Hypothesis

As mentioned above, in order to improve the power-saving benefit of BLI, we proposed a layered embedded engine (LEE) concept, in which the freestream and the boundary layer flow are ingested separately. The difference in configuration between LEE and conventional embedded engine is shown in Figure 1. In the previous work, a double-diffuser S-shaped intake was designed and compared with a single-diffuser S-shaped intake to preliminarily examine its internal flow features. For these two intakes, the global parameters length and overall offset are kept the same. In the simulation, the back pressure at the outlets are adjusted so that the sum of mass flow $\dot{m}$ of the two diffusers of the double-diffuser S-shaped intake is the same as that of the single-diffuser S-shaped intake under the same inlet conditions (the Mach Number of freestream and the boundary layer thickness are the same). Figure 2 shows the comparison of flow pattern between double-diffuser S-shaped intake and single-diffuser S-shaped intake. The performance of these two intakes such as mass flow $\dot{m}$, total pressure recovery (TPR) and distortion coefficient (DPCP$_{avg}$) at the aerodynamic interface plane (AIP, which is referred to the outlet of the S-shaped diffuser in the present study) are also listed (detailed definition of TPR and DPCP$_{avg}$ can be found in Section 3.3). For the double-diffuser S-shaped intake, the subscript fs denotes the diffuser which ingests the freestream and the subscript bl denotes the diffuser which ingests the boundary layer flow. It is worth noting that when the boundary layer flow is ingested into the lower S-shaped diffuser which has a smaller spacing between the inner and outer walls, the flow separation is greatly weakened compared to the single-diffuser S-shaped intake, resulting in a significant reduction in the low total pressure area at AIP. The total pressure recovery TPR of the lower S-shaped inlet which only ingests the boundary layer flow is increased by 4.5% compared to the single-diffuser S-shaped intake, and distortion index DPCP$_{avg}$ is decreased by 53.2%.

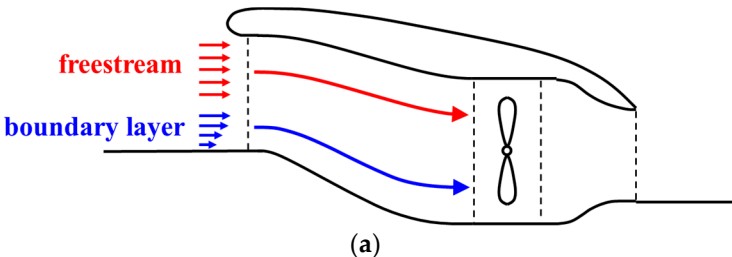

(**a**)

**Figure 1.** *Cont.*

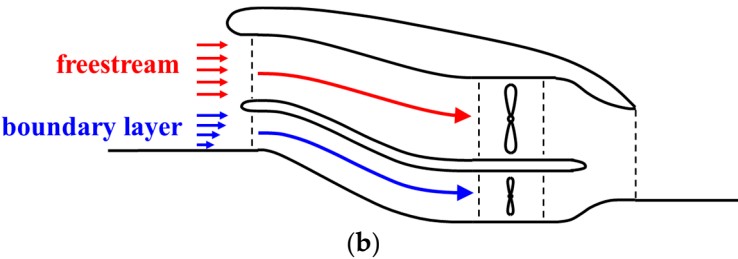

**Figure 1.** Comparison of conventional embedded engine and layered embedded engine (LEE): (**a**) conventional embedded engine; (**b**) layered embedded engine (LEE).

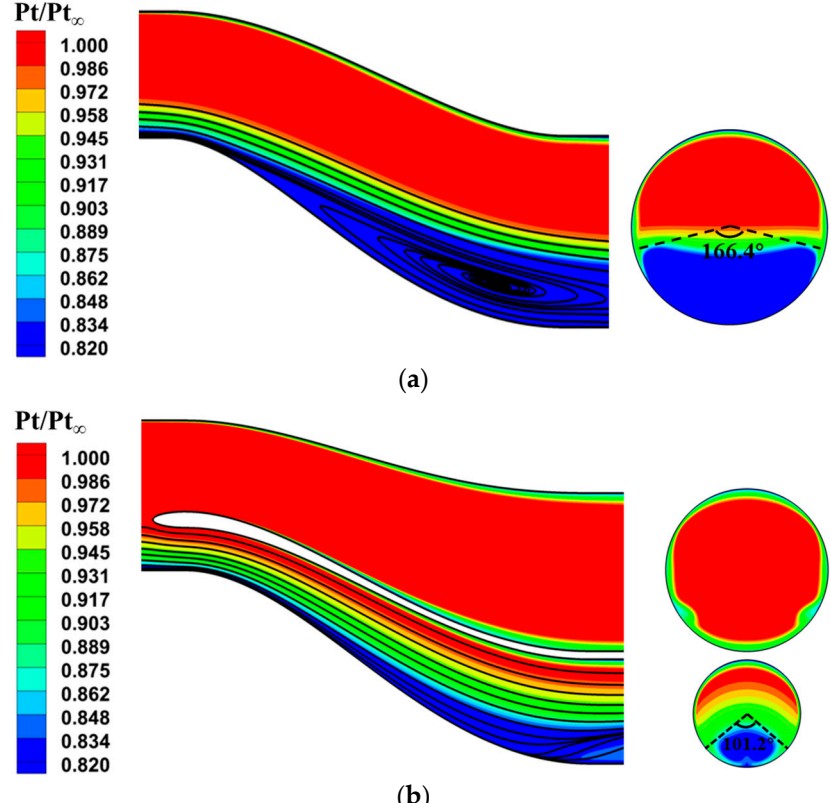

**Figure 2.** Comparison of flow pattern between double-diffuser S-shaped intake and single-diffuser S-shaped intake. Total pressure contour and boundary layer streamlines at symmetry plane (left), total pressure contour at AIP (right). (**a**) Single-diffuser S-shaped intake: $\dot{m}$ = 176.2 kg/s, TPR = 0.9368, DPCP$_{avg}$ = 0.0998; (**b**) double-diffuser S-shaped intake: $\dot{m}_{fs}$ = 126.06 kg/s, TPR$_{fs}$ = 0.9924, DPCP$_{avg,fs}$ = 0.0008; $\dot{m}_{bl}$ = 50.70 kg/s, TPR$_{bl}$ = 0.9818, DPCP$_{avg,bl}$ = 0.0467.

The above phenomenon shows that, for a BWB aircraft with embedded engine configuration, modifying the engine configuration to ingest the freestream and the boundary layer flow separately can reduce the loss due to flow separation to improve intake performance. In addition, it can be expected that the efficiency of downstream fan will also be improved due to reduced distortion at AIP. In order to understand the reason for this phenomenon, we reviewed the mechanism of the flow separation within the S-shaped diffuser.

Figure 3 shows the typical wall static pressure distribution in an S-shaped diffuser. Previous studies have shown that the reason for the flow separation in an S-shaped diffuser is the adverse pressure gradient of the inner wall which is mainly due to two reasons. On the one hand, the diffusion of flow in the diffuser generates the adverse pressure gradient along the flow direction which is the same as the diffusion of flow in a straight duct, the degree of which depends on the relative difference between the inlet and outlet Mach numbers. On the other hand, the normal pressure gradient provides the centripetal force

required by the flow turning in two bends of opposite curvature. The flow near the inner wall at the first bend creates a negative pressure zone due to the convex curvature, while the flow near the inner wall at second bend creates a positive pressure zone, eventually forming the aggressive adverse pressure gradient on the inner wall along the flow direction [26]. In addition, the flow turning at two bends can also form the adverse pressure gradient on the inner wall even though there is no diffusion in the S-shaped duct (see Figure 3b). For the lower S-shaped diffuser which only ingests the boundary layer flow, since the average Mach number of the incoming boundary layer is closer to the fan face Mach number compared to the freestream, the boundary layer flow does not require a large degree of diffusion. Therefore, it should be the flow turning but not the flow diffusion that affects its internal flow separation features.

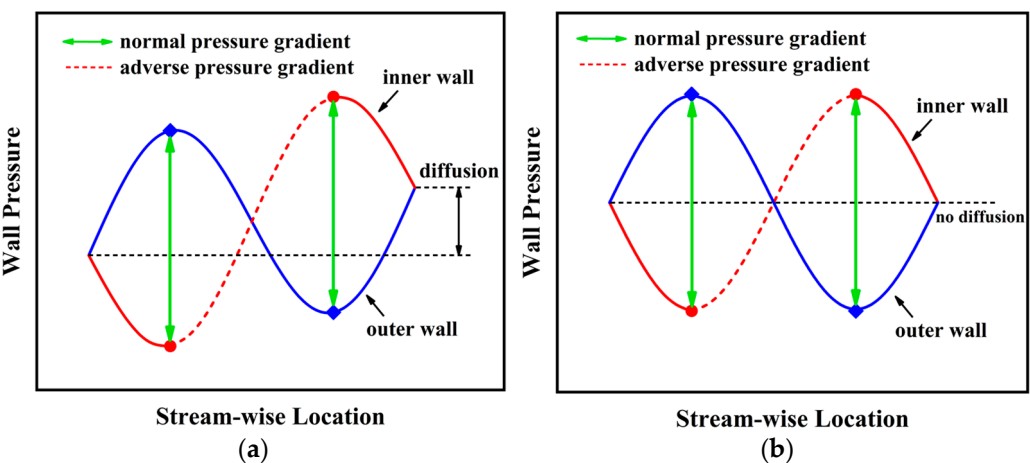

**Figure 3.** Typical wall static pressure distribution in the S-shaped diffuser. (**a**) S-shaped diffuser with diffusion; (**b**) S-shaped duct without diffusion.

In order to analyze the effect of wall pressure distribution on flow separation features from the second aspect above, the key geometric parameters affecting the normal pressure gradient of the S-shaped diffuser are further explored and correlated with the geometric characteristics of the lower S-shaped diffuser which only ingests the boundary layer flow. The analysis is performed in a standard 2D S-shaped duct of equal height which walls is simplified to arcs and the two bends are centrosymmetric as shown in Figure 4. Here, the equal height means that the distance between the upper and lower walls is constant along the flow direction. The purpose of setting this assumption is to exclude the influence of flow diffusion on the analysis of pressure distribution.

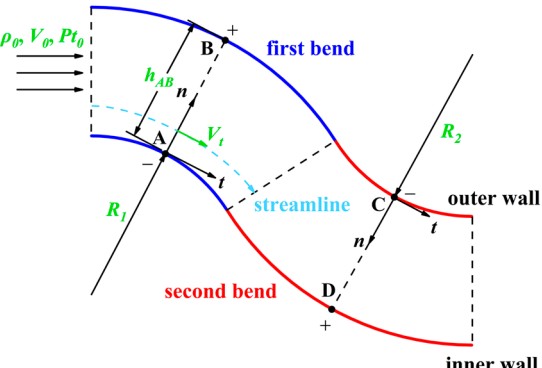

**Figure 4.** Ideal flow in a 2D S-shaped duct.

Assume that a uniform flow with density $\rho_0$, velocity $V_0$ and total pressure $Pt_0$ enters into the duct in the horizontal direction. The flow within the duct is assumed to be steady,

inviscid, incompressible, and the streamlines are parallel to each other in the two bends. These assumptions mean that the fluid inside the duct has no normal velocity component in the natural coordinate system. Then, the normal momentum equation in the natural coordinate system is given as:

$$\frac{1}{\rho_0}\frac{\partial p}{\partial n} = \frac{V_t^2}{R} \tag{1}$$

where $p$ is the static pressure, $n$ is the normal coordinate in the natural coordinate system which is perpendicular to local velocity, $V_t$ is the tangential velocity component in the natural coordinate system and $R$ is the local curvature radius. Taking the above formula at a fixed flow direction position, such as point A, gives:

$$\frac{1}{\rho_0}\frac{dp}{dn} = \frac{V_t^2}{R_1} \tag{2}$$

where $R_1$ is the curvature radius of the inner wall at first bend. Under the assumptions mentioned above, the Bernoulli equation is hold on different streamlines:

$$\frac{1}{2}\rho_0 V_t^2 + p = Pt_0 \tag{3}$$

for uniform inlet condition and inviscid flow, the total pressure $Pt_0$ remains the same on different streamlines. Substituting Equation (3) into Equation (2) and integrate from point A to point B gives:

$$\left(1 + \frac{h_{AB}}{R_1}\right)^2 = \frac{Pt_0 - P_A}{Pt_0 - P_B} \tag{4}$$

where $h_{AB}$ is the normal distance between point A and point B. Since the selection of point A in the derivation of Equation (4) is arbitrary, the above formula is valid for any flow position in a standard 2D S-shaped duct, which gives:

$$\left(1 + \frac{h_i}{R_i}\right)^2 = \frac{Pt_0 - P_{i,inner}}{Pt_0 - P_{i,outer}} \tag{5}$$

where $i$ is an arbitrary flow position. Equation (5) gives the relation between the pressure difference between the inner and outer walls and the local height-to-radius ratio (HRR) $h_i/R_i$ at any flow direction in 2D S-shaped duct. If the wall curvature does not change significantly, the left side of Equation (4) keeps approaching one when $h_i$ decreases, which means decreasing the difference between the inner and outer wall pressure. Suppose that point A is just taken at the lowest point of the inner wall pressure and point D is just taken as the highest point of the inner wall pressure, the normal pressure gradient mitigates as $h_i/R_i$ decreases, which intuitively means that the pressure distribution shape in Figure 3 becomes flatter, so that the adverse pressure gradient on the inner wall decreases.

The above derivation gives a hypothetical explanation for the aforementioned flow phenomena occurring in the lower S-shaped diffuser which only ingests the boundary layer flow, and the purpose of this paper is to further verify this hypothesis using numerical methods. It should be noted that HRR is a local parameter. It can be seen from the above derivation that the condition that HRR is the same along the flow direction is that the curvature of the duct wall is the same everywhere and the area ratio is 1. For a 3D S-shaped diffuser, the above conditions are difficult to guarantee, because the wall is usually not a circular arc and the area ratio of the diffuser is usually greater than 1. In order to make the parameter HRR more general to characterize the normal pressure gradient in the S-shaped diffuser. In the subsequent studies, the parameter HRR takes the form given in Equation (6), where $H_{in}$ is the inlet height of S-shaped diffuser and $L/\Delta H$ is the length-to-offset ratio. The reason for this replacement is that, for the design of an S-shaped diffuser, the distance between the inner and outer wall is usually related to the design mass flow rate. Therefore, the inlet height can globally reflect the average distance of the inner and outer walls. In

addition, the wall curvature of the S-shaped diffuser is usually different at different flow positions, but the degree of the S-shaped diffuser curvature is globally related to the length and offset. Therefore, the global parameter $L/\Delta H$ which is usually used in the S-shaped diffuser design can reflect the average curvature change of the duct wall.

$$\text{HRR} = \frac{H_{in}}{\frac{1}{4}\Delta H\left[1 + \left(\frac{L}{\Delta H}\right)^2\right]} \tag{6}$$

## 3. Methodology

### 3.1. S-Shaped Diffuser Geometry

The design method of S-shaped diffuser developed by Wellborn et al. [13] has a good ability to describe the geometric characteristics of the S-shaped diffuser, and their research provides good experimental data to validate the numerical method. Thus, this paper adopts their method to construct the S-shaped diffuser geometry. The geometry of the S-shaped diffuser is shown in Figure 5. The centerline consists of two arcs with the same radius $R$ and central angle $\theta_{max}/2$ but face opposite directions. The radius $R$ and central angle $\theta_{max}/2$ can be determined by the global parameter length $L$ and offset $\Delta H$. All cross sections perpendicular to the centerline are circular. The variation of the duct radius as a function of the angle $\theta$ is describe by cubic polynomial; for more details see Wellborn et al. [13]. In the present study, $H_{in}$ for determining HRR is equal to inlet diameter of the S-shaped diffuser.

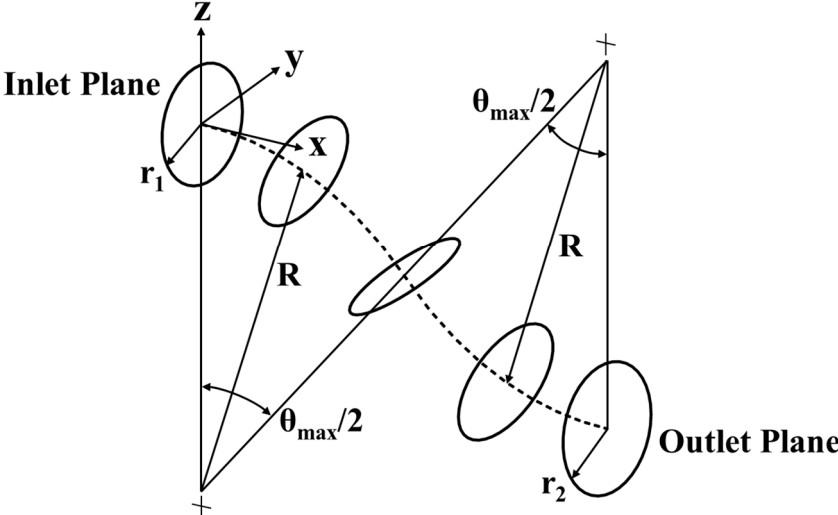

**Figure 5.** Schematic showing S-shaped diffuser geometry.

### 3.2. Numerical Methods

In the present study, the 3D hybrid combination of the finite volume method and finite element method CFD commercial solve, ANSYS CFX 18.2, is applied to compute the flow field in the S-shaped diffuser. The 3D steady Reynolds-averaged Navier-Stokes (RANS) equations are selected to depict the conservation of mass, momentum and energy of the air ideal gas. The two-equation $k$-$\omega$ Shear-Stress-Transport (SST) model is selected to simulate the turbulent flow, which has been found to provide good accuracy for S-shaped diffuser internal flow analysis [27,28]. To improve the accuracy of the simulations, the second-order upwind scheme is selected to discrete the advection and turbulence terms. The timescale factor is set as 2 to ensure the robustness and speed of the solution. The convergence criterion is set to a value of $10^{-6}$ for the RMS residual values.

For duct-only internal flow simulation, in order to accurately simulate boundary layer ingestion, the turbulent flow over a 3D flat plate is first solved. The results at the location of desired boundary layer thickness of this simulation are then used to extract total pressure, total temperature and turbulent profiles (turbulent kinetic energy and turbulent eddy

frequency), which are applied to the mesh nodes on the inlet face of the S-shaped diffuser. The average static pressure is specified at the outlet of the S-shaped diffuser, and the value is adjusted to fit the common fan-face Mach Number (usually around 0.55) but not the design mass flow since the inlet height of the S-shaped diffuser of all test cases are various. Solid walls are defined as nonslip and adiabatic.

The 3D structured meshes are topologically generated by the commercial software ANSYS ICEM, as shown in Figure 6. The inlet and outlet planes are both extended to $5r_2$ to improve iteration convergence according to Laplace equation. A multi-block structured O-topology grid is used to define the computational domain. In particular, the first level grid height is set as 0.003 mm within the boundary layer, and the growth rate is 1.2. The maximum y+ adjacent to the solid walls are less than 2. The number of mesh cells is about 1.2 million.

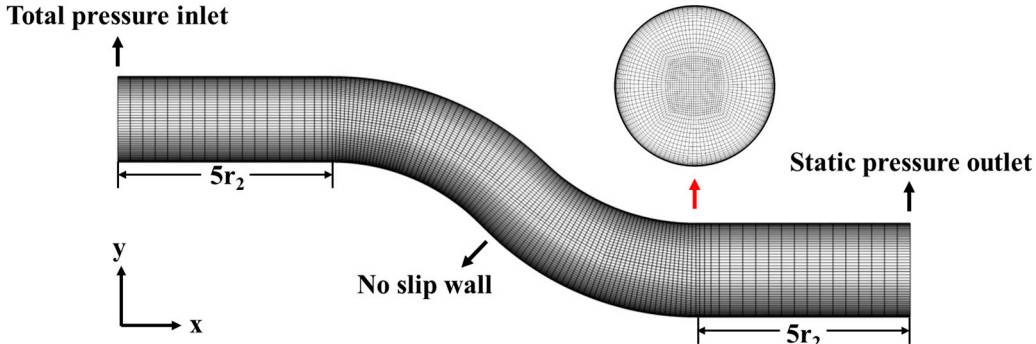

**Figure 6.** Mesh system and computational domain for S-shaped diffuser.

*3.3. Validation*

For the numerical study, the first important consideration is to guarantee the mesh independency. Therefore, a set of grids sequentially named as coarse-, fine- and dense-mesh were checked with an inlet centerline Mach number of 0.6 and a Reynolds number (base on the inlet centerline velocity and inlet diameter) of $2.6 \times 10^6$. Table 1 tabulates the number of mesh cells, the calculated boundary layer profile parameters and the absolute error of the boundary layer profile parameters relative to the experimental data of by Wellborn et al. [13] (data in brackets). The boundary layer thickness $\delta$, the displacement thickness $\delta_1$ and the momentum thickness $\delta_2$ are calculated from the boundary layer velocity profile of the flow field one inlet radius upstream of the inlet plane, and all parameters are normalized by the inlet radius $r_1$. Meanwhile, the boundary layer shape factor $H$, which is defined as the ratio of $\delta_1$ to $\delta_2$, is also calculated. Comparisons indicate that enhancing the mesh resolution from coarse mesh to fine mesh has greatly reduced the prediction error; however, when the mesh resolution is further enhanced to dense mesh, the prediction error changes little.

**Table 1.** Comparison of inlet boundary layer parameters with different-sized meshes.

| Data Source | Number of Mesh Cells | $(\delta/r_1) \times 100$ | $(\delta_1/r_1) \times 100$ | $(\delta_2/r_1) \times 100$ | $H$ |
|---|---|---|---|---|---|
| Experiment [13] | - | 6.95 (−) | 1.46 (−) | 1.06 (−) | 1.38 (−) |
| Coarse mesh | 0.57 million | 6.603 (4.99%) | 1.37 (6.16%) | 1.019 (3.87%) | 1.344 (2.61%) |
| Fine mesh | 1.20 million | 6.931 (0.28%) | 1.423 (2.53%) | 1.026 (3.21%) | 1.387 (0.51%) |
| Dense mesh | 2.53 million | 6.933 (0.24%) | 1.424 (2.47%) | 1.027 (3.11%) | 1.387 (0.51%) |

Figure 7 shows the comparison of wall static pressure coefficient distributions along the axial direction for two circumferential positions with different-sized meshes. The wall static pressure near the inlet and outlet plane predicted by different-sized meshes remains consistent with the experimental data [13]. However, the separation point predicted by the coarse mesh is closer to the downstream relative to the experimental data [13], resulting in a large deviation in wall static pressure of 170° circumferential position from $s/d$ = 2.17

to *s/d* = 4.17, while the wall static pressure after the separation point predicted by the fine and dense meshes is in better agreement with the experimental data [13]. Considering the computational efficiency and accuracy for the later simulations, the fine mesh is selected for all the subsequent test cases.

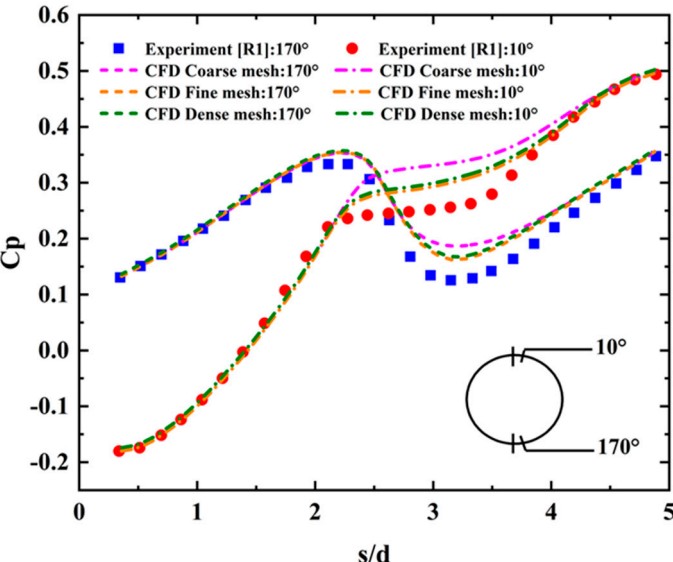

**Figure 7.** Comparison of wall static pressure coefficient distributions along the axial direction for two circumferential positions with different-sized meshes.

Diffuser performance is typically evaluated based on the total pressure recovery ratio that describes the losses in the duct. Total pressure recovery is defined as:

$$\text{TPR} = \frac{Pt_{avg,AIP}}{Pt_{avg,inlet}} \tag{7}$$

where $Pt_{avg,inlet}$ is the average total pressure at inlet and $Pt_{avg,AIP}$ is the average total pressure at AIP.

Flow distortion at AIP is also significant since it can negatively impact fan performance. The circumferential distortion is evaluated based on $\text{DPCP}_{avg}$ (average circumferential distortion descriptor) by using the SAE recommended practices [29]. $\text{DPCP}_{avg}$ is defined as:

$$\text{DPCP}_{avg} = \frac{1}{m} \sum_{i=1}^{m} \left[ \frac{Pt_{avg,i} - Pt_{low,avg,i}}{Pt_{avg,i}} \right] \tag{8}$$

The AIP is partitioned into $i = 1 \ldots m$ rings where $i$ is the ring number on the AIP rake, $Pt_{avg,i}$ is the average total pressure of ring $i$, and $Pt_{low,avg,i}$ is the average total pressure in the region where the total pressure is below $Pt_{avg,i}$.

The static pressure coefficient $C_p$ is used to discuss the static pressure distribution on the wall of the S-shaped diffuser, which is defined as:

$$C_p = \frac{p - p_\infty}{q_\infty} \tag{9}$$

where $p$ and $p_\infty$ are the static pressures on the wall surface of the S-shaped diffuser and of the freestream air far from the inlet plane, respectively. $q_\infty$ is the dynamic pressure of the freestream air.

*3.4. Test Cases*

Test conditions for S-shaped diffusers are listed in Table 2. In the present study, the length to offset ratio (LOR) and area ratio (AR) of the S-shaped diffuser for all test cases is set as 2.41 and 1.2. Since the variation of HRR is achieved by changing the inlet height, the Reynolds number is calculated with the characteristic length of the inlet height of the S-shaped diffuser, and the Reynolds number is within the self-modeling zone in all test cases. In addition, the average Mach number at AIP is kept around 0.5 for all test cases by adjusting the back pressure in the numerical simulations. The purpose of these practices is to ensure the similarity of the flow field within the S-shaped diffuser since the geometry and inlet condition are different for all test cases.

**Table 2.** Test conditions for S-shaped diffusers.

| Test Case | HRR | BLI (%) | $\text{Re}_{\text{Hin}} \times 10^{-6}$ | $\text{Ma}_{\text{avg,AIP}}$ |
|:---:|:---:|:---:|:---:|:---:|
| Case 1 | 0.20 | 0 | 3.62 | 0.496 |
| Case 2 | 0.20 | 33 | 3.51 | 0.494 |
| Case 3 | 0.20 | 66 | 3.32 | 0.493 |
| Case 4 | 0.20 | 100 | 3.13 | 0.491 |
| Case 5 | 0.25 | 0 | 4.57 | 0.501 |
| Case 6 | 0.25 | 80 | 4.08 | 0.497 |
| Case 7 | 0.33 | 0 | 6.15 | 0.503 |
| Case 8 | 0.33 | 60 | 5.65 | 0.498 |
| Case 9 | 0.50 | 0 | 9.24 | 0.506 |
| Case 10 | 0.50 | 33 | 8.80 | 0.505 |
| Case 11 | 0.50 | 40 | 8.71 | 0.505 |
| Case 12 | 0.50 | 66 | 8.33 | 0.502 |
| Case 13 | 0.50 | 100 | 7.82 | 0.499 |

## 4. Results

*4.1. Verification under Uniform Inlet Condition*

Figure 8 shows the normalized velocity contour and streamlines near the inner wall on the symmetry plane in S-shaped diffuser with various HRR under uniform inlet conditions. For different HRR, the reverse flow and the low-velocity area indicate that different degrees of flow separation occur on the inner wall at the first bend. To observe the flow separation region more clearly, the limiting streamlines and wall shear stress on the inner wall of the S-shaped diffuser are extracted as shown in Figure 9. For different HRR, the structure of the flow separation is basically the same. The boundary layer on the inner wall near the symmetry plane first separated at first bend. The limiting streamlines of the boundary layer near the sidewall converge toward the inner wall and reattach at the second bend, eventually forming a pair of symmetrical separation vortices. However, the range of the flow separation region is greatly reduced when HRR is decreased from 0.50 to 0.20. Figure 10 shows the comparison of flow separation features with various HRR under uniform inlet condition, where $X_{sp}/L$ and $X_{rp}/L$ denotes the $X$ coordinate of the separation point and the reattachment point normalized by the length of the duct and $L_s/L$ denotes the relative range of the flow separation region which is calculated by subtracting $X_{sp}/L$ from $X_{rp}/L$. When HRR decreases from 0.50 to 0.20, the relative position of the separation point $X_{sp}/L$ moves downstream and the relative position of the reattachment point $X_{rp}/L$ moves upstream, resulting in a great reduction in the relative range of the flow separation region $L_s/L$ from 34.0% to 4.0%.

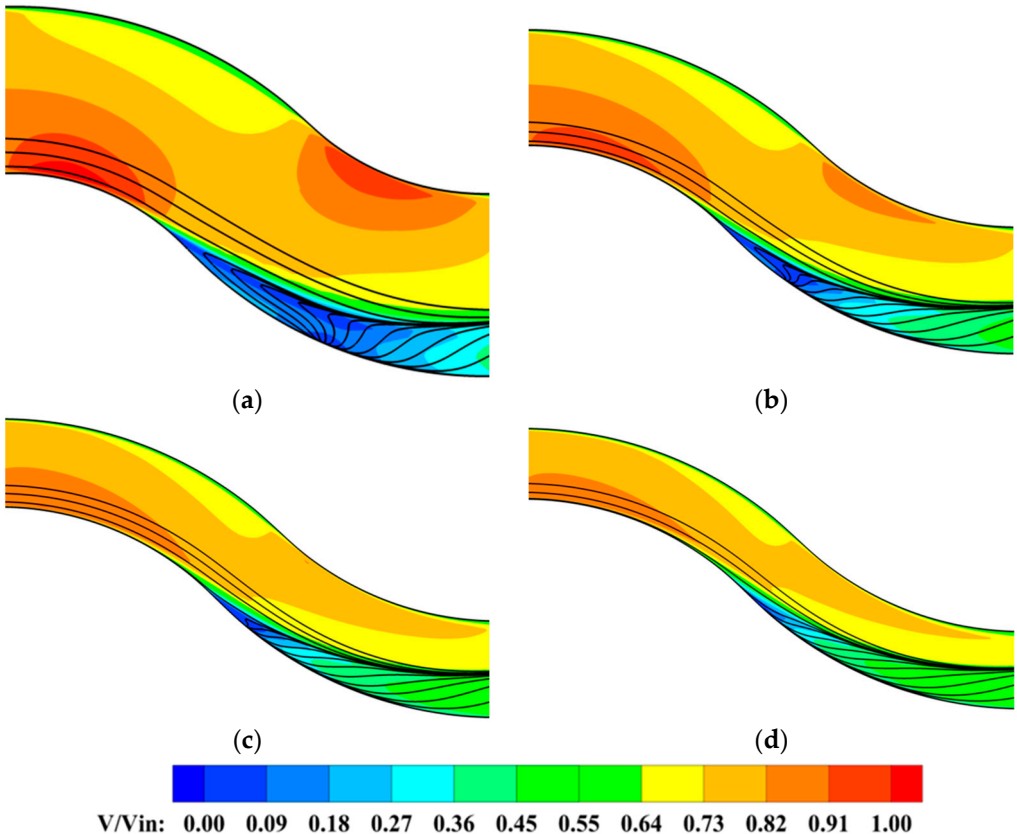

**Figure 8.** Normalized velocity contour and streamlines near inner wall on symmetry plane in S-shaped diffuser with various HRR under uniform inlet condition: (**a**) Case 9: HRR = 0.50, BLI = 0%; (**b**) Case 7: HRR = 0.33, BLI = 0%; (**c**) Case 5: HRR = 0.25, BLI = 0%; (**d**) Case 1: HRR = 0.20, BLI = 0%.

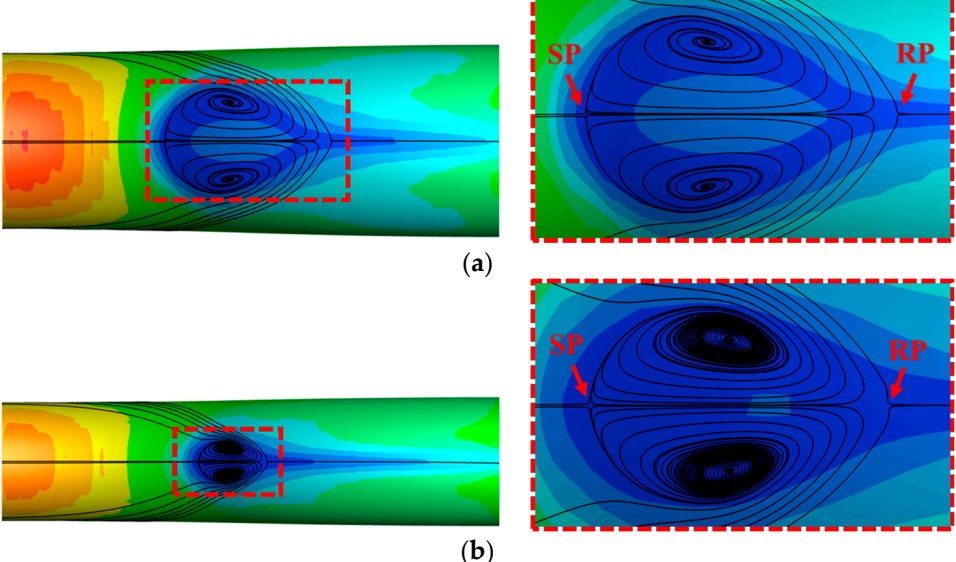

**Figure 9.** *Cont.*

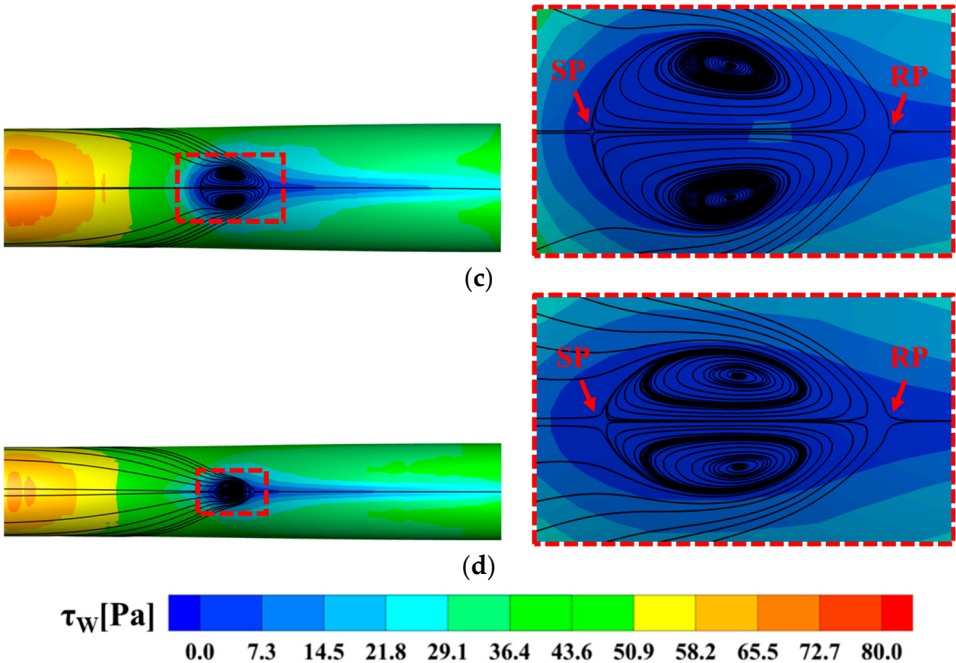

**Figure 9.** Limiting streamlines and wall shear stress on inner wall of S-shaped diffuser with various HRR under uniform inlet condition (the right side is an enlargement of the flow separation region which is highlighted in the red dashed box. SP denotes the separation point and RP denotes the reattachment point): (**a**) Case 9: HRR = 0.50, BLI = 0%; (**b**) Case 7: HRR = 0.33, BLI = 0%; (**c**) Case 5: HRR = 0.25, BLI = 0%; (**d**) Case 1: HRR = 0.20, BLI = 0%.

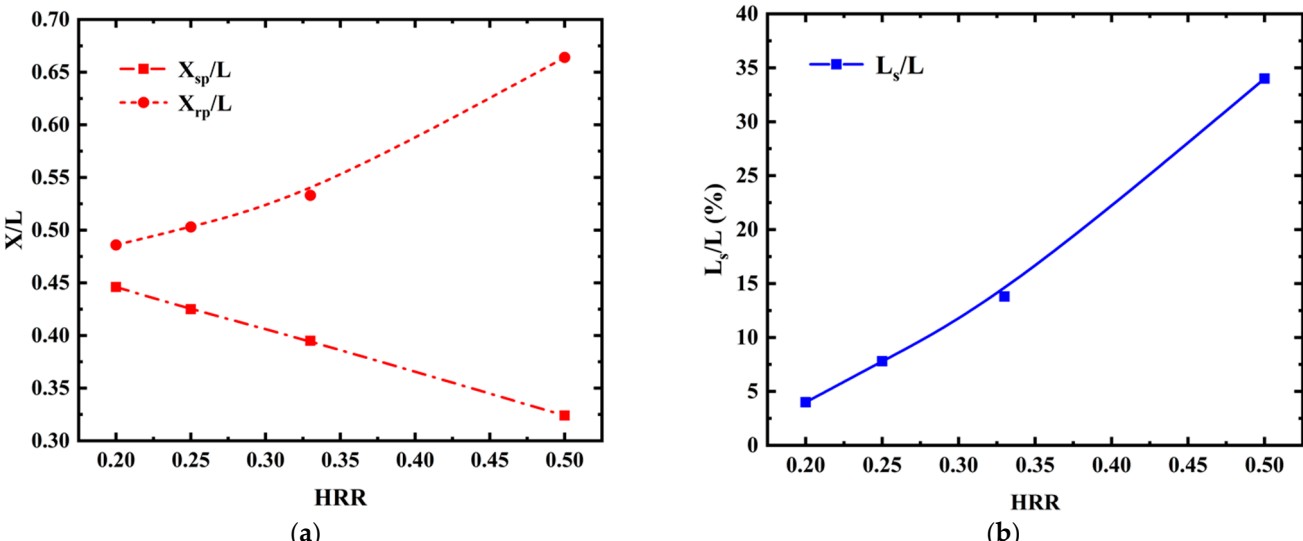

**Figure 10.** Comparison of flow separation features with various HRR under uniform inlet condition: (**a**) relative position of the separation and reattachment point; (**b**) relative range of the flow separation region.

The above arguments indicate that the parameter HRR proposed in Section 2 indeed affects the flow separation features within the S-shaped diffuser under uniform inlet conditions. In order to confirm whether this effect is caused by the change in pressure gradient, the static pressure distribution of the inner and outer walls of the S-shaped diffuser with different HRR under uniform inlet condition is extracted as shown in Figure 11. It can be seen that as HRR decreases, the pressure gradient between the inner and outer walls at the first bend and second bend decreases, which is precisely due to the decrease in the centripetal force required by the flow to turn. Therefore, the adverse pressure gradient

on the inner wall at the first bend which causes the flow separation decreased. This is consistent with the analysis of the effect of HRR on the normal pressure gradient in Section 2. However, the differences are that this decrease in the normal pressure gradient is due to the increase in the minimum static pressure on the inner wall at first bend and on the outer wall at second bend, while the static pressure on the outer wall at first bend and on the inner wall at second bend seem to have little change. This shows that the change in the normal pressure gradient caused by HRR only acts on a single wall, rather than uniformly acting on both inner and outer walls as discussed in Section 2. In addition, the flow separation region can also be identified from the trend of static pressure distribution in Figure 11. After a period of adverse pressure gradient on the inner wall at first bend, the inflection point of the static pressure change indicates the occurrence of the flow separation. In the flow separation region, the static pressure first remains essentially unchanged and then begins to rise as the reattachment point is approached. In general, the decrease in HRR does reduce the normal pressure gradient of the flow at first bend and second bend, which in turn reduces the adverse pressure gradient on the inner wall at first bend, eventually causing the downstream movement of the separation point and a great reduction in the range of the flow separation region.

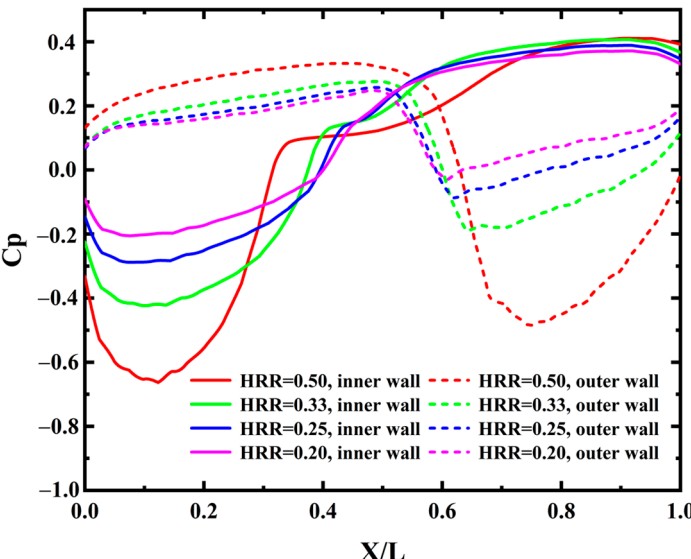

**Figure 11.** Comparison of static pressure distribution of inner and outer wall of S-shaped diffuser with various HRR under uniform inlet condition.

### 4.2. Effect of HRR on Separation Features with BLI

The above discussions prove that under uniform inlet conditions, the decrease in HRR of the S-shaped diffuser can reduce the normal pressure gradient required by the flow to turn, thereby reducing the adverse pressure gradient on the inner wall at the first bend, eventually causing a great reduction in the range of the flow separation region. An immediate question is whether such conclusions hold in the presence of BLI. In order to study the effect of HRR on flow separation features in the S-shaped diffuser with BLI, two cases with HRR = 0.50 and HRR = 0.20 were selected, and simulations were carried out under the conditions of the inlet relative height of BLI being 0%, 33%, 66% and 100%.

Figure 12 shows the limiting streamlines near the inner wall of the S-shaped diffuser with various inlet BLI for different HRR. For all test cases, the structure of the flow separation all appear as a pair of symmetrical separation vortices, which is the same as the uniform inlet condition, indicating that changing inlet condition does not change the flow separation structure. Figure 13 shows the comparison of flow separation features with various inlet BLIs for different HRRs. It is worth noting that, no matter in the cases of HRR = 0.50 or 0.20, when the inlet condition changes from uniform to BLI, the flow separation features show a sudden change. For a fixed HRR, the separation point moves upstream and

the reattachment point moves downstream when the inlet condition changes from uniform (BLI = 0%) to BLI = 33%, resulting in an extension of the flow separation region. Taking HRR = 0.50 as an example, when the inlet condition changes from BLI = 0% to 33%, the relative position of the separation point $X_{sp}/L$ moves upstream of 0.03 and the relative position of the reattachment point $X_{rp}/L$ moves downstream of 0.045, resulting in an extension in the range of the flow separation region $L_s/L$ from 34.0% to 41.5%. However, when the inlet relative height of BLI further increases from 33% to 100%, the position of separation and reattachment point changes little, and so does the range of the flow separation region. For a fixed inlet BLI, the separation point moves downstream and the reattachment point moves upstream when HRR is reduced from 0.50 to 0.20, resulting in a great reduction in the range of the flow separation region. Taking BLI = 100% as an example, when HRR decreases from 0.50 to 0.20, the relative position of the separation point $X_{sp}/L$ moves downstream of 0.136 and the reattachment point $X_{rp}/L$ moves upstream of 0.207, resulting in a great reduction in the range of the flow separation region $L_S/L$ from 42.8% to 8.5%. This shows that the flow separation features within the S-shaped diffuser are very sensitive to the change in HRR but not to the change in the inlet relative height of BLI.

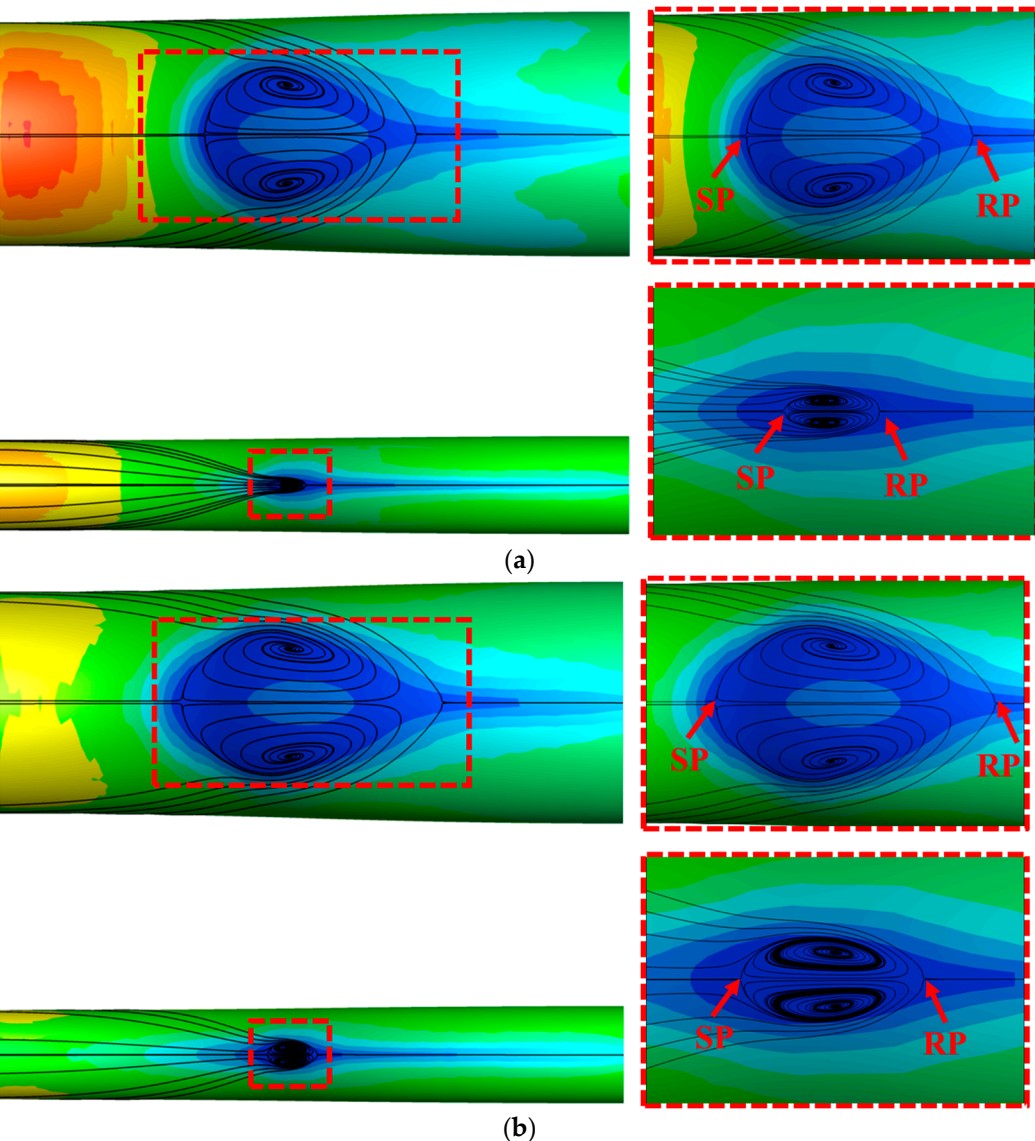

**Figure 12.** *Cont.*

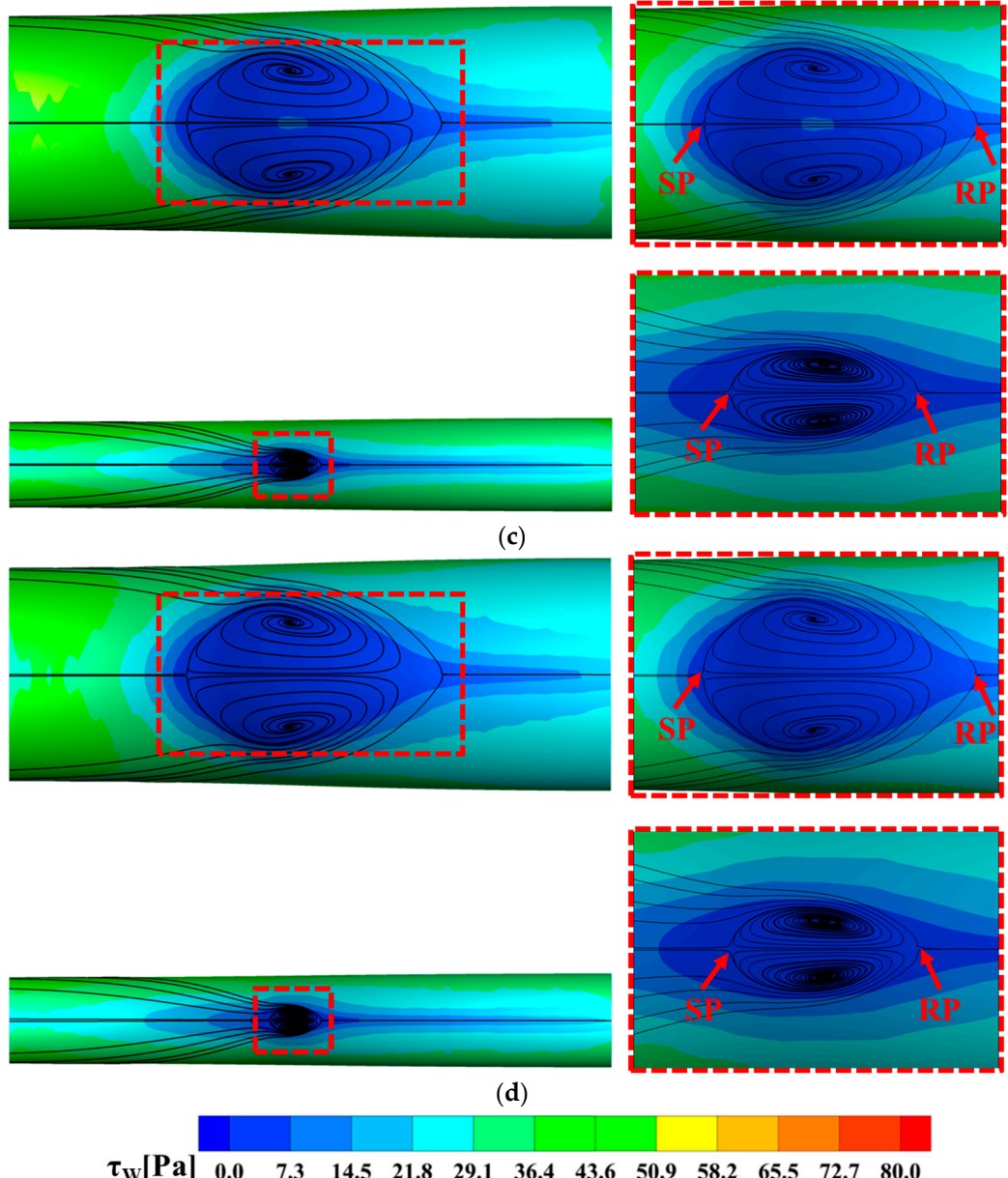

**Figure 12.** Limiting streamlines and wall shear stress near inner wall of S-shaped diffuser with various inlet BLIs for different HRRs (the right side is an enlargement of the flow separation region which is highlighted in the red dashed box. SP denotes the separation point and RP denotes the reattachment point): (**a**) top: Case 9, HRR = 0.50, BLI = 0%; bottom: Case 1, HRR = 0.20, BLI = 0%; (**b**) top: Case 10, HRR = 0.50, BLI = 33%; bottom: Case 2, HRR = 0.20, BLI = 33%; (**c**) top: Case 12, HRR = 0.50, BLI = 66%; bottom: Case 3, HRR = 0.20, BLI = 66%; (**d**) top: Case 13, HRR = 0.50, BLI = 100%; bottom: Case 4, HRR = 0.20, BLI = 100%.

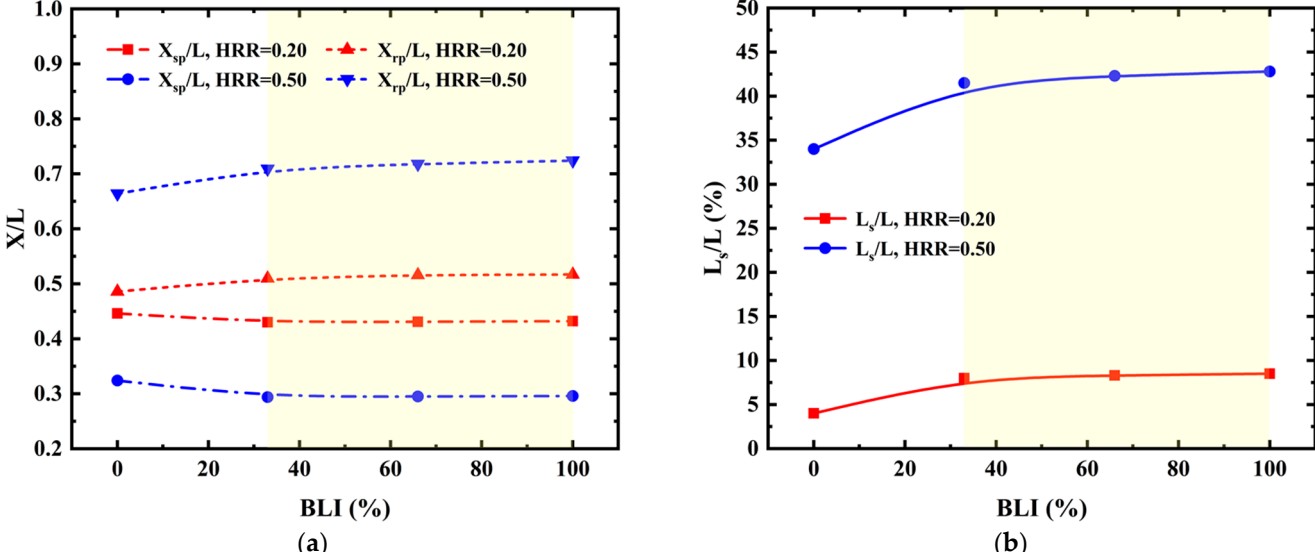

(a)                                              (b)

**Figure 13.** Comparison of flow separation features with various inlet BLIs for different HRRs: (**a**) relative position of the separation and reattachment point; (**b**) relative range of the flow separation region.

In order to verify whether the effect of HRR on flow separation features in the S-shaped diffuser with BLI is caused by the pressure gradient, the static pressure distribution of inner and outer walls of the S-shaped diffuser with various inlet BLI for different HRR is shown in Figure 14. The position of the separation point and the range of the flow separation region can be identified through the trend of the static pressure change on the inner wall, in which it is easier to see the influence of HRR on the flow separation features. It can be clearly seen from Figure 14 that, under different inlet BLIs, the normal pressure gradient required by the flow to turn at first bend and second bend decreases as HRR decreases, which in turn reduce the adverse pressure gradient on the inner wall at the first bend, and eventually the range of the flow separation region is reduced. The above discussions show that the conclusions drawn under uniform inlet conditions are still valid for BLI. In addition, by comparing the four cases with the same inlet conditions but different HRRs, it can also be seen that the flow separation features in S-shaped diffuser are sensitive to the change in HRR, but not to the change in the relative height of BLI.

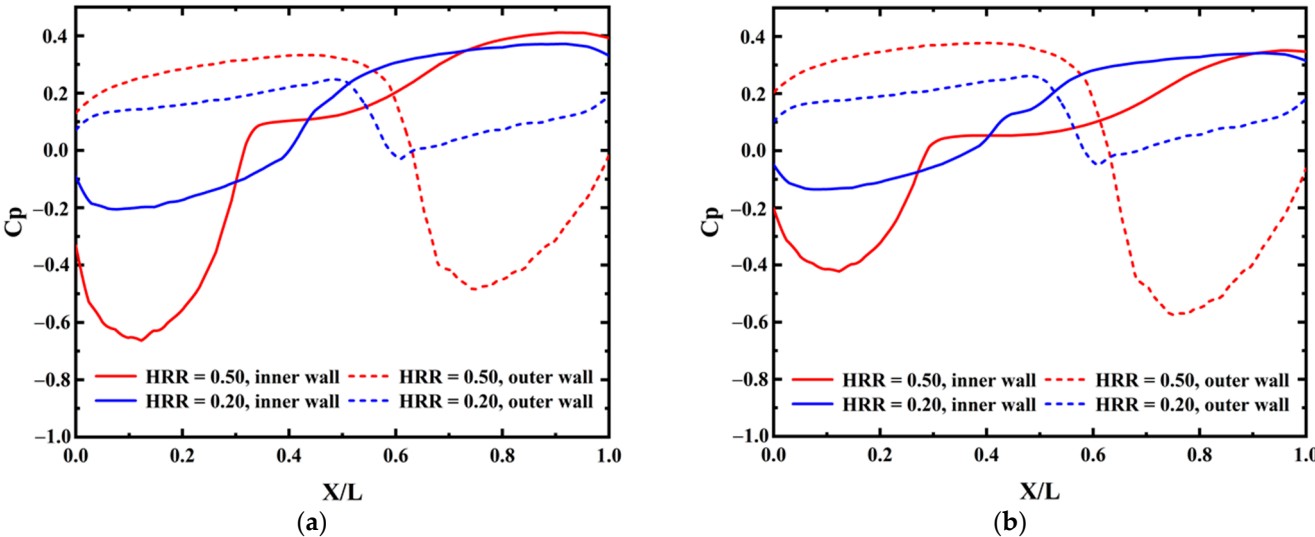

(a)                                              (b)

**Figure 14.** *Cont.*

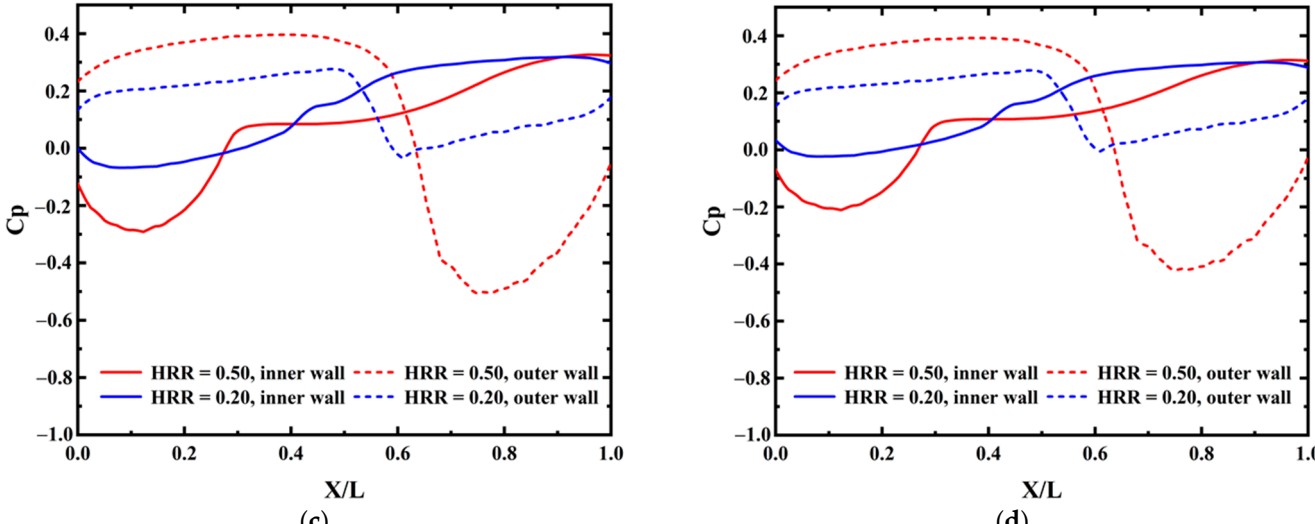

**Figure 14.** Comparison of static pressure distribution of inner and outer wall of S-shaped diffuser with various inlet BLIs for different HRRs: (**a**) BLI = 0%; (**b**) BLI = 33%; (**c**) BLI = 66%; (**d**) BLI = 100%.

*4.3. Application under Fixed Boundary Layer Thickness*

The above discussions show that reducing HRR with various inlet BLI can effectively weaken the flow separation in the S-shaped diffuser. However, for the above cases, the absolute height of BLI at the inlet of the S-shaped diffuser is different even though the relative height of BLI is the same due to different HRRs. In an actual situation, when the installation position of the embedded engine on the BWB aircraft is determined, the state of the boundary layer is mainly related to the development distance. That is to say, the absolute height of the boundary layer is basically constant when the boundary layer flow reaches the inlet of the embedded engine. For current BWB aircraft using embedded engine configuration, the thrust requirements of the aircraft determine the design mass flow rate of the engine, which in turn determines the inlet height of the embedded engine, and this height is usually greater than the height of the local boundary layer (the relative height of BLI is about 20% to 40% according to different BWB aircraft design). According to the above conclusions, when the length-to-offset ratio is constant, a larger inlet height means a larger HRR, which may aggravate the risk of flow separation of the boundary layer flow in the S-shaped diffuser. Therefore, in order to reduce the losses caused by the flow separation of the boundary layer flow in the S-shaped diffuser, a feasible way is to reduce the inlet height as much as possible to only ingest the boundary layer flow. In order to verify this feasibility, the inlet absolute height of BLI of the S-shaped diffuser is the same in the following four cases, while the HRR is gradually reduced, resulting in the inlet relative height of BLI with 40%, 60%, 80% and 100%.

Figure 15 shows the limiting streamlines and wall shear stress near the inner wall of the S-shaped diffuser with various HRRs under same absolute height of inlet BLI. It can be seen that when the absolute height of inlet BLI remains constant, reducing HRR can delay the position of the separation point and reduce the range of the flow separation region. Figure 16 shows the comparison of flow separation features with various HRRs under the same absolute height of inlet BLI. When HRR decreases from 0.50 to 0.20, the relative position of the separation point $X_{sp}/L$ moves downstream from 0.293 to 0.432, and the range of the flow separation region $L_s/L$ greatly reduces from 41.9% to 8.5%. This confirms the conclusions drawn in the above study that the flow separation features in the S-shaped diffuser are very sensitive to the change in HRR but not to the change in the inlet relative height of BLI. Similar to the analysis in Section 4.2, the effect of HRR on the flow separation features can also be explained by the static pressure distribution on the inner and outer walls as shown in Figure 17, so the details will not be repeated here.

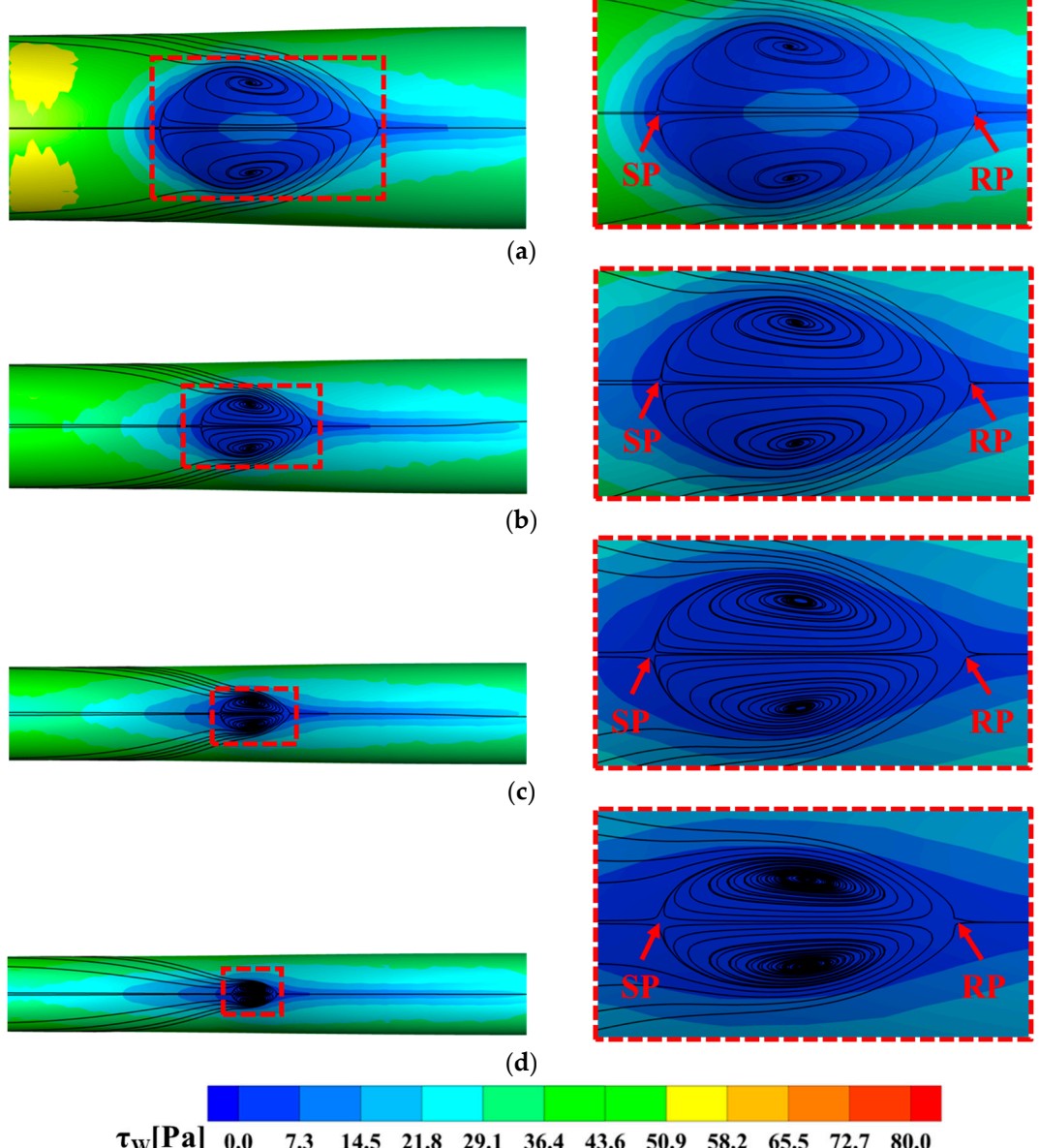

**Figure 15.** Limiting streamlines and wall shear stress near inner wall of the S-shaped diffuser with various HRRs under same absolute height of inlet BLI (the right side is an enlargement of the flow separation region which is highlighted in the red dashed box. SP denotes the separation point and RP denotes the reattachment point): (**a**) Case 11: HRR = 0.50, BLI = 40%; (**b**) Case 8: HRR = 0.33, BLI = 60%; (**c**) Case 6: HRR = 0.25, BLI = 80%; (**d**) Case 4: HRR = 0.20, BLI = 100%.

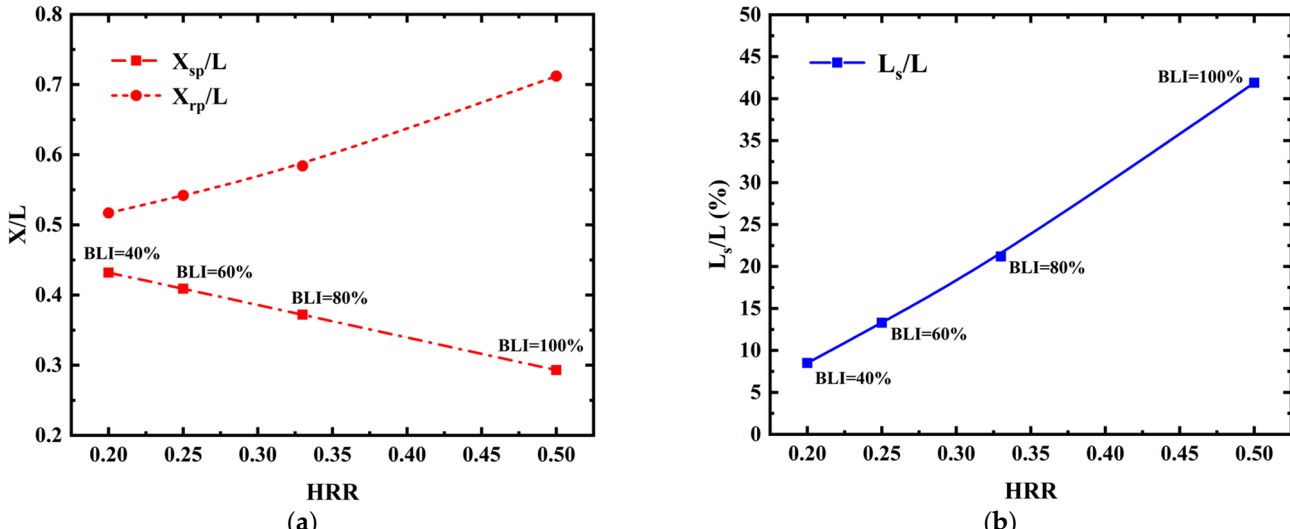

**Figure 16.** Comparison of flow separation features with various HRR under same absolute height of inlet BLI: (**a**) relative position of the separation and reattachment point; (**b**) relative range of the flow separation region.

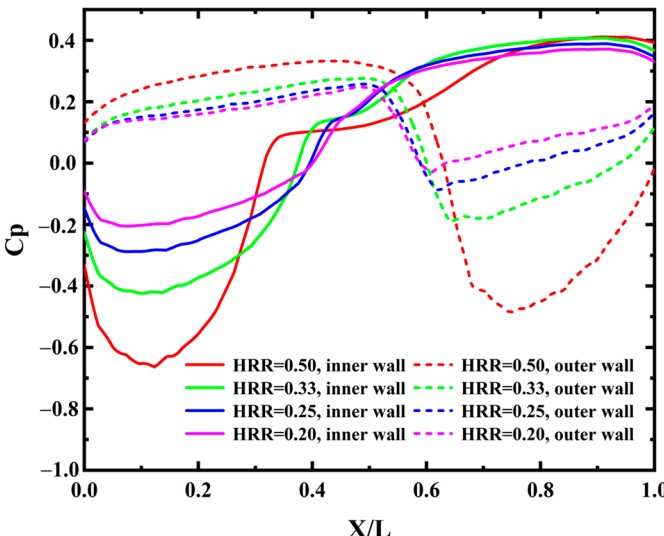

**Figure 17.** Comparison of static pressure distribution of the inner and outer wall of S-shaped diffuser with various HRRs under same absolute height of inlet BLI.

The above discussions show that when the absolute height of inlet BLI is fixed due to the aircraft design, the flow separation within the S-shaped diffuser can be effectively weakened by reducing the design HRR to only ingest the boundary layer flow. For these four designs with different HRRs, the performance of the S-shaped diffuser is of great interest. Figure 18 shows the normalized total pressure distribution at inlet and AIP of the S-shaped diffuser with different designs of HRR but same absolute height of inlet BLI, and the circumferential distortion coefficient $DPCP_{avg}$ at inlet and AIP of these four cases are also calculated. The reason that $DPCP_{avg}$ at inlet station is also calculated is to examine the development of the distortion caused by BLI after passing through the S-shaped diffuser. It can be seen that when the absolute height of inlet BLI is the same, the $DPCP_{avg}$ at inlet is basically the same, except the $DPCP_{avg}$ with HRR = 0.50 has a deviation of about 0.005 from the other three cases. Therefore, it can be seen that when the absolute height of inlet BLI is the same, the distortion degree at the inlet of S-shaped diffuser with different design HRR is basically the same. However, the low total pressure region at AIP

decreases significantly as HRR decreases due to the suppressed flow separation, resulting in a large decrease in DPCP$_{avg}$ at AIP. When HRR is reduced from 0.50 to 0.20, DPCP$_{avg}$ at AIP decreases by about 53.3% and TPR increases about 0.7% as shown in Figure 19. The above discussions show that for a fixed boundary layer height, the flow separation can be effectively suppressed by designing the S-shaped diffuser with a smaller HRR, so that the inherent distortion at inlet of embedded engine due to BLI will not be significantly magnified after the low-momentum flow passing through the S-shaped diffuser. It can be expected that the efficiency and stability of the downstream fan can also be improved due to the substantial reduction in distortion at AIP.

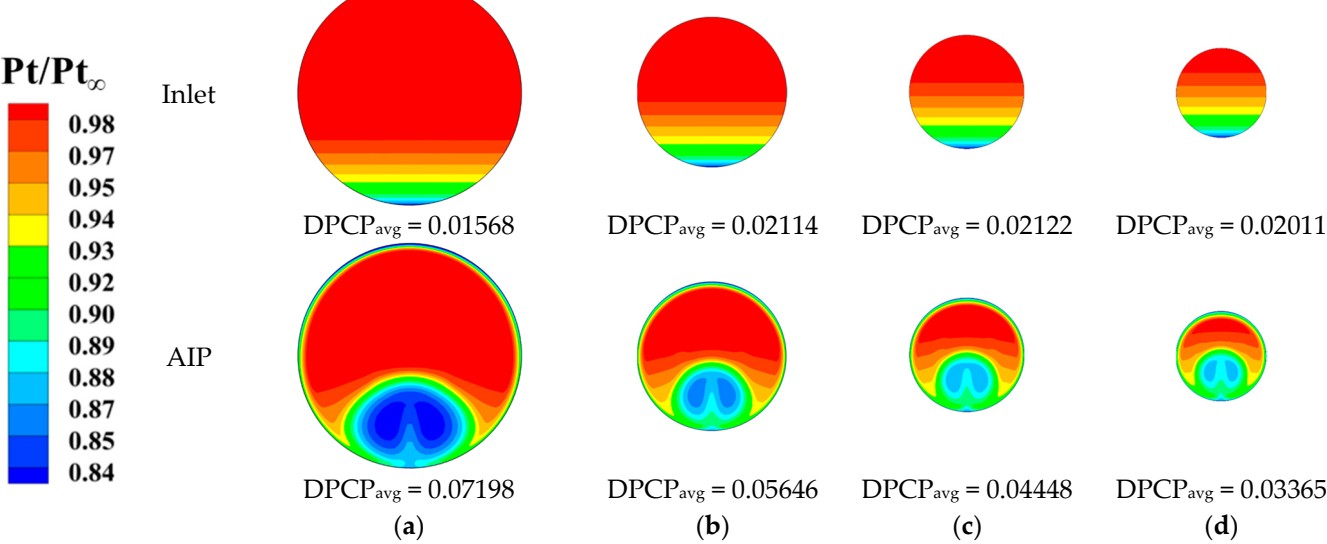

**Figure 18.** Comparison of normalized total pressure distribution at inlet and AIP of S-shaped diffuser with various HRR under same absolute height of inlet BLI: (**a**) Case 11: HRR = 0.50, BLI = 40%; (**b**) Case 8: HRR = 0.33, BLI = 60%; (**c**) Case 6: HRR = 0.25, BLI = 80%; (**d**) Case 4: HRR = 0.20, BLI = 100%.

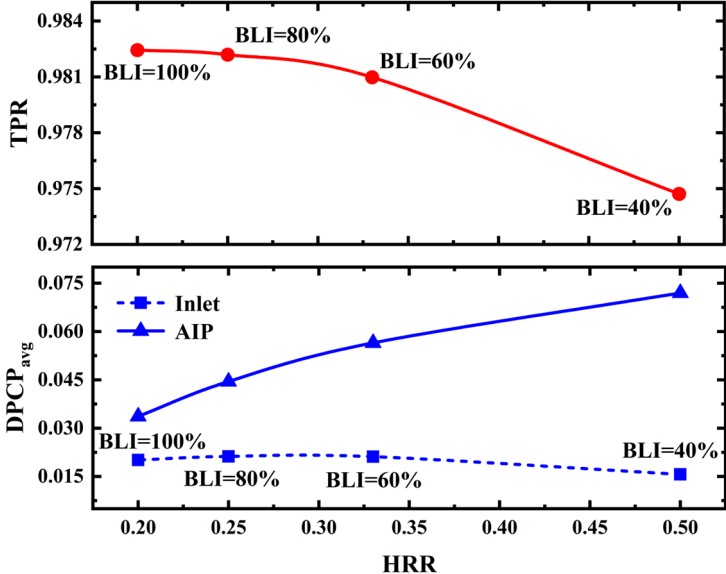

**Figure 19.** Total pressure recovery TPR and circumferential distortion coefficient DPCP$_{avg}$ as a function of HRR with same absolute height of inlet BLI.

## 5. Conclusions

Previous studies have found that when the boundary layer flow is completely ingested into an S-shaped diffuser with a small distance between the inner and outer walls, the internal flow separation is significantly suppressed. This phenomenon motivated us to conduct the present research. First, a hypothesis that the parameter height-to-radius ratio (HRR) may have significant effect on the flow separation features in the S-shaped diffuser was proposed. Second, numerical methods were used to verify this hypothesis and the effect of HRR on flow separation features in the S-shaped diffuser with BLI are further investigated. Finally, the possible applications of the findings of this study in practical design are discussed. The conclusions are summarized as follows:

(1) Whether the inlet condition is uniform or BLI, the normal pressure gradient required by the flow to turn at first and second bends in the S-shaped diffuser is decreased when HRR decreases, and thus the adverse pressure gradient along the flow direction on the inner wall at first bend is weakened, resulting in suppressed flow separation. In addition, it was also found that the flow separation features in the S-shaped diffuser are very sensitive to the change in HRR but not to the change in relative height of inlet BLI. This conclusion provides perspective for designing a low-loss S-shaped diffuser under inherent inlet distortion caused by BLI.

(2) When the absolute height of inlet BLI is the same, reducing HRR does significantly suppress the flow separation within the S-shaped diffuser, and the inherent distortion due to BLI at inlet station is not amplified after the low-momentum boundary layer flow passing through the diffuser, resulting in lower distortion at AIP and higher total pressure recovery. The results therefore indicate that, for the BWB aircraft with embedded engine configuration, by designing the intake with smaller HRR to only ingest the boundary layer flow can greatly improve the performance of the S-shaped diffuser. It can be expected that the efficiency and stability of the downstream fan can also be improved due to the reduced distortion at AIP.

**Author Contributions:** Conceptualization, Z.L., T.P. and Y.L.; methodology, validation, formal analysis and investigation, Y.L. and Y.Z.; resources, Z.L. and T.P.; data curation, Z.L. and Y.L.; writing—original draft preparation, Y.L.; writing—review and editing, Z.L., T.P. and Y.Z.; supervision and funding acquisition, Z.L. and T.P. All authors have read and agreed to the published version of the manuscript.

**Funding:** The authors acknowledge the support of National Natural Science Foundation of China (Nos. 52176032 and 51976005), ScienceCenter for Gas Turbine Project (P2022-B-II-004-001), Advanced Jet Propulsion Creativity Center, AEAC (Project ID. HKCX2020-02-013), National Science and Technology Major Project (2017-II-0005-0018), the Foundation of National Key Laboratory of Science and Technology on Aero-Engine Aero-thermodynamics (WDZC2019601A202), the Fundamental Research Funds for the Central Universities, and Beijing Nova Program.

**Data Availability Statement:** The data used to support the findings of this study are available from the corresponding author upon request.

**Conflicts of Interest:** The authors declare no conflict of interest.

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
