# Peer review of "Numerical Investigation on the Effect of Height-to-Radius Ratio on Flow Separation Features in S-Shaped Diffuser with Boundary Layer Ingestion"

_aerospace, doi:10.3390/aerospace10060551_

Round 1
Reviewer 1 Report
In this manuscript authors attempted to investigate the impact of the height-to-radius ratio (HRR) on flow separation features in an S-shaped diffuser with boundary layer ingestion (BLI). Authors proposed hypothesis in this study, suggested that HRR could also significantly influence flow separation in the S-shaped diffuser. To examine this hypothesis, numerical methods were employed. Initially, the study confirmed that HRR does indeed affect flow separation features in the S-shaped diffuser under uniform inlet conditions. Subsequently, the influence of HRR on flow separation was examined under different inlet relative heights of BLI. Interestingly, the results indicated that changes in HRR had a notable impact on flow separation, whereas alterations in the inlet relative height of BLI had less effect. Also, when considering a fixed boundary layer height generated from the airframe, it was observed that a smaller design HRR in the S-shaped diffuser effectively mitigated flow separation. This suppression led to higher total pressure recovery and reduced distortion coefficients, thereby enhancing the aerodynamic performance of the embedded engine with BLI.
I appreciate the authors effort to simulate the flow mechanism of S-shaped diffuser with boundary layer ingestion. The findings provide valuable insights for optimizing the aerodynamic performance of engines.
I am providing my questions below
1. Was there Mesh independent study conducted? If so please provide the details.
2. If possible please provide kinetic energy variation along the diffuser.
3. I understand that commercial package CFX is used, please provide more numerical background for the simulation.
4. How did the flow separation features in the S-shaped diffuser respond to changes in HRR and the inlet relative height of BLI?
5. How does reducing the HRR parameter affect flow separation and pressure gradients in the S-shaped diffuser under both uniform and BLI inlet conditions?
Reviewer 2 Report
The authors present a numerical study of flow S-shaped diffusers. The presentation of results and their discussion provides some interesting points for others in the field, and overall the work merits publication, provided the following points are addressed:
- you give definitions for the three-letter acronyms (TLA's) used in the article, apart from AIP - please add
- there is no discussion of mesh convergence in the article: as this is a numerical study, this is a key aspect of the method, and this needs to be addressed, in particular, repeat runs with mesh size halved and quartered should be performed and the results documented, in particular, how do the velocity profiles (away from the inlet) and pressure distribution along the way change with increased mesh density
- lines 174-5 - it seems as though you are excluding diffusion, yet in Figure 3, that seems to be the main effect - please explain
- the derivation from lines 172-205 seems to be for a plug flow entering the duct, yet it is asserted in the next paragraph that the analysis is suitable for flows with a boundary layer profile, that is not a plug profile - this part needs to be carefully modified to explain why the (essentially) inviscid profile result can be used for studying a viscous profile case.
- symbols are sometimes shown in italics, and sometimes shown without italics - please be consistent
- contour plots - the numerical values in the colour bars are too small - please increase their size so they are easy to read
- it is asserted (line350 et seq) that where "the static pressure remains relatively flat is the separated region" - yet on the subsequent figures (eg Fig. 10), the regions marked as having relatively flat pressure do not in fact have a flat distribution along that length: for example, HRR=0.5, inner wall, Cp increases from about 0.1 to more than 0.2, more than doubling. Either: reconsider this definition and use the actually flat pressure length, or, (re)define what is meant by relatively flat - this is a weakness that needs to be resolved.
- the results shown in the contour plots and Cp-x line graphs are summarized in tables, but I think it would be more useful to consolidate and plot the key results, so that relative effects can be assessed. I was pleased to see Fig. 16 in this regard, and think that graphical, as opposed to tabular, summaries would greatly add to this workl
In general, the article is well written, but there are still some idiosyncrasies that need careful attention. I would suggest getting an English writer or a journal editing service to go through the text.
Round 2
Reviewer 2 Report
The revised version dramatically improves the quality of the paper. Well done to the authors.
Minor edits would aid readability.